# Veteran trees have divergent effects on beetle diversity and wood decomposition

**Ross Wetherbee** *, **Tone Birkemoe**, **Ryan C. Burner**, **Anne Sverdrup-Thygeson**

Faculty of Environmental Sciences and Natural Resource Management, Norwegian University of Life Sciences, Ås, Norway

* ross.wetherbee@nmbu.no

## Abstract

Veteran hollow trees are keystone structures in ecosystems and provide important habitat for a diverse set of organisms, many of which are involved in the process of decomposition. Since veteran trees are 'islands' of high biodiversity, they provide a unique system in which to study the relationship between biodiversity and decomposition of wood. We tested this relationship with a balanced experiential design, where we quantified the taxonomic and functional diversity of beetles directly involved in the process of decomposing wood, and measured the decomposition of experimentally added bundles of small diameter wood around 20 veteran trees and 20 nearby young trees in southern Norway. We found that the diversity (both taxonomic and functional) of wood-decomposing beetles was significantly higher around the veteran trees, and beetle communities around veteran trees consisted of species with a greater preference for larger diameter wood. We extracted few beetles from the experimentally added wood bundles, regardless of the tree type that they were placed near, but decomposition rates were significantly lower around veteran trees. We speculate that slower decomposition rates around veteran trees could have been a result of a greater diversity of competing fungi, which has been found to decrease decay rates. Veteran trees provide an ecological legacy within anthropogenic landscapes, enhance biodiversity and influence wood decomposition. Actions to protect veteran trees are urgently needed in order to save these valuable organisms and their associated biodiversity.

## Introduction

Veteran hollow trees are valuable entities in forests, farmlands, traditional landscapes and urban areas because they are keystone structures that increase habitat heterogeneity and biodiversity [1–3]. However, they are declining globally [4]. The decline of veteran trees is adding to the existential threat of global biodiversity loss, which is especially problematic because biodiversity contributes to critical ecosystem functions on which humans rely [5]. Research clearly indicates that loss of biodiversity results in reductions in these contributions [6, 7], but the exact relationship between biodiversity and ecosystem functioning continues to be debated [8–11].

A community's contribution to ecosystem functioning is more closely related to its diversity of relevant functional traits than to the number of species within the community [12–14]. A

**Data Availability Statement:** All species observations related to this study are registered in Global Biodiversity Information Facility (GBIF) and are publicly available (https://doi.org/10.15468/5bxyph), and all data that was used in this study

has been archived in NMBU Open Research Data
(https://dataverse.no/dataverse/nmbu).

**Funding:** The author(s) received no specific
funding for this work.

**Competing interests:** The authors have declared
that no competing interests exist.

diversity of functional traits in a community is thought to promote multiple ecosystem processes and make these processes more resilient to change [15–19]. Veteran trees are, in a sense, 'islands' of high biodiversity, as they are surrounded by other trees that support less species-rich communities [1]. This provides the opportunity for paired study designs that examine ecosystem functioning at different levels of diversity within the same landscape context [20]. Thus, research related to the influence of veteran trees, and their associated biodiversity, on ecosystem functioning has a two-fold advantage, in that it can elucidate the relationship between biodiversity and ecosystem function and simultaneously provide incentives to protect these valuable organisms.

Dead wood is a particularly important form of plant biomass, as there are more than 73 billion tons of carbon stored in naturally occurring dead wood globally [21]. Small diameter wood in particular may be of great importance regarding carbon cycling and storage in forest ecosystems [22]. However, biodiversity mediates the rate at which wood decomposes and this in turn effects the rate at which carbon dioxide is released to the atmosphere [23]. Veteran trees have larger circumference, a greater diversity of dead wood and more fungal fruiting bodies than typical forests trees [2], and this increases the diversity of wood-decomposing organisms associated with them [24–26]. However, the relationship between biodiversity and decomposition is complex, with many interactions among diverse organisms [11, 27].

Insects are extraordinarily diverse, interact with many organisms involved in decomposition [28, 29], and likely play an important role in the process of decomposing wood [30]. In northern ecosystems, where termites are absent, beetles are one of the primary insect decomposers [26, 31] and both beetle species richness and abundance are often high in small diameter wood [32–35]. Beetles contribute to decomposition both directly, by consuming dead wood and the fungi living within it [36–39], and indirectly, most likely by acting as dispersers of fungal spores to dead wood [40, 41]. These indirect effects are especially relevant, because fungi are a primary driver of wood decomposition [42, 43], and decomposition rates have been shown to decrease in response to insect exclusion [41]. It is therefore likely that the diversity of organisms associated with veteran trees has some effect on decomposition of wood, yet this remains relatively unexplored.

Combining an analysis of functional diversity with a study exploring the relationships between biodiversity and decomposition of wood may help to expand the current understanding of that relationship. For example, the body size of an insect is tightly connected to its resource use [14] and, in the case of wood-boring beetles, also influences how tunnels and galleries within the wood are created [38, 39]. Additionally, features of the dead wood, such as the diameter, decay stage, vertical position within the canopy and tree species, have been found to influence insect communities and decomposition rates [25, 32, 33, 44, 45]. Therefore, a diversity of niche preferences within an insect community may decrease competition and increase resource partitioning [46–48], thus potentially increasing decomposition rates [8].

In order to test the influence of veteran trees on wood-decomposing beetle diversity and wood decomposition rates, we employed a paired experimental design. We matched veteran oaks with nearby young oaks in southern Norway, experimentally added bundles of recently cut small diameter oak branches, and sampled beetle communities with window traps over a two-year period. We measured the diversity of wood-decomposing beetle communities and quantified wood decay rates. The study had four aims: 1) measure the number of beetles species associated with wood decomposition around veteran and young trees, 2) compare the two communities' functional diversity, 3) investigate whether this diversity increased the number of beetles colonizing the experimentally added wood, and 4) measure the decay rates of the wood bundles. We predicted that there would be a greater diversity of wood-decomposing beetles around veteran trees, in terms of both number of species and functional diversity, and

that this would result in more beetles colonizing the wood bundles, which in turn would increase decomposition rates.

## Methods

This study complied with the appropriate institutional, national, and international guidelines and was approved by the Norwegian University of Life Sciences, Faculty of Environmental Sciences and Natural Resource Management (Project number: 7101212). The study sties were on public and private land, and we confirm that Vestfold Fylkeskommune and the private land-owners provided permission to conduct the study at these sites.

We established a paired experimental design, in which we randomly chose twenty veteran oaks in the central distribution of oaks in Norway from the Norwegian database of veteran oaks [49]. Each of these trees had a circumference of 2 m or greater (measured at the height of 130 cm). We subsequently matched each veteran tree with a young oak that was within 200 m and had similar immediate surroundings (e.g. similar openness, sun exposure and surrounding tree species). The tree-pairs were in either forests or open landscapes (n = 12 and 8, respectively) and were within a 30 km radius of the city of Larvik. The higher number of tree-pairs in the forest was due to the difficulty of finding suitable young oaks near the veteran oaks in open landscapes. A young tree was defined as an oak that had a circumference less than 150 cm and no visible hollow. The tree-pairs were always more than 100 meters apart, but 12 of these pairs were clustered within four 500 m x 500 m sampling blocks that were established by the survey which originally identified the veteran oaks [49]. These same tree-pairs were also used by Wetherbee et al., (2020b). The mean circumference of the veteran oaks was 283 cm (200–405 cm) and mean circumference of the young oaks was 74.5 cm (25–148 cm).

We measured beetle diversity, and decomposition rates of experimentally added wood, from spring 2017 to fall 2018. To measure beetle diversity, we hung one flight intercept trap in the canopy of each tree. The flight intercept traps were made of two intersecting 20 x 40 cm windows with a funnel below leading to a vial containing propylene glycol, water (4:1 mixture) and a drop of detergent used as a surfactant. They were hung from a branch in the canopy, on the opposite side of the tree as the experimentally added wood. The traps were active from May to August in both years and emptied once a month during that time.

In order to measure colonization and decomposition of small diameter wood, we transported a total of 149.5 kg of fresh oak branches into the forest. The branches were divided evenly into bundles that were held together with zip ties, and two bundles were placed near each tree (one on the ground at the base of the tree and one hanging in the mid-canopy, 3–4 m high). The wood bundles consisted of six 50 cm long freshly cut branches that were 1–3 cm in dimeter, and the average wet weight of the bundles was 1.9 kg (min = 1.4, max = 2.6). All branches originated from three living oaks and were collected on the 9[th] of May 2017. Subsequently, the branches were transported back to the lab, where they were randomly mixed, sorted into bundles and weighed. The bundles were then transported to the field sites between the 16[th] and the 25[th] of May 2017. Bundles were retrieved between the 13[th] and the 18[th] of August 2018.

After the bundles were collected from the field, they were placed directly into rearing chambers. The rearing chambers consisted of non-transparent cardboard barrels (50 cm diameter and 150 cm length) with a plastic lid and a transparent collection vial mounted on it. The bundles were then reared for one year in an open-air building at the ambient temperature (September 2018 to October 2019). The collection vials were emptied regularly and at the end of rearing all invertebrates found inside the barrels were collected. Subsequently, the bundles were weighed, then oven dried at $103^0$ C until the weight stabilized (approximately 7 days)

before measuring dry mass. In order to measure bundle densities, we cut off 5 cm from both ends of the sticks (from here on referred to as 'tips') before and after the field experiment. All tips were oven dried at $103^0$ C until the weight had stabilized (approximately 4 days). The tips were then weighed, and the volume was measured by water displacement. The density of the tips was then calculated as the dry mass divided by the volume [41].

All beetles collected in both the flight intercept traps and in the bundle extractions were identified to the species level by an expert, following the taxonomy of The Norwegian Biodiversity Information Centre [50]. Additionally, all species observations were registered in Global Biodiversity Information Facility (GBIF) and are publicly available [51]. Following the protocol set by Wetherbee et al. (2020a), a literature survey was carried out and beetles were classified as wood-decomposers if they were described as xylophagous, mycetophagous (fungivore) or saprophagous and polyphagous (including a life stage that feeds on wood either directly or indirectly) at any point in their life stages (S1 Table). Available trait information that was relevant for wood decomposition rates was collected from the literature (Table 1).

Functional diversity was subsequently calculated based on all traits, and a community weighted mean (CWM) was calculated for each trait individually. We verified that at least 80% of all species used in the statistical analysis had trait information, since functional diversity indices are sensitive to missing trait information [52]. All species that were excluded due to a lack of trait data were rare in our data set (total abundance < 3). We chose to use functional dispersion (FDis) as our measure of functional diversity because it accounts for species abundances, can be calculated for multiple traits, and is only minimally sensitive to species richness [53]. FDis is a measure of dispersion in trait space and is calculated as the mean distance of all species (weighted by abundances) to the centroid of the community in multidimensional trait space [53]. We also choose to use CWMs of each trait to gain insight into how mean trait values differed between veteran and young oaks. CWM is defined as the mean values of a given trait present in the community, weighted by the relative abundance of the taxa bearing each trait value [54].

All analyses were carried out in R version 3.4.0 [55]. Species richness and FDis of saproxylic beetles, as well as CWM for each trait, were calculated with the *dbFD* function in the 'FD' package. FDis was calculated using a Gower dissimilarity matrix and the "cailliez" correction method [53, 56]. Additionally, the percent weight loss (PWL) of the dead wood bundles was calculated as the start weight minus the end weight, divided by the start weight.

In order to test the effect of the tree type on beetle diversity and decomposition rates, we fit a set of models to predict beetle species richness, FDis, CWM for each trait, and the density and PWL of the bundles. Although we were primarily focused on the dichotomy between the veteran and the young trees, we also included relevant tree and landscape predictor variables and, in the case of the wood bundles, location (hanging or on the ground) as fixed effects in the models (Table 2, S1 Fig). We determined that the response variables of density of the bundles, FDis and CWM for each trait were all approximately normally distributed and modeled

**Table 1. Traits included in our measure of wood-decomposing beetle functional diversity (for species list and trait values see S1 Table).**

| Trait | Link to decomposition | Type | Collection source |
|---|---|---|---|
| Body length | Closely linked to many life history traits such as life span and dispersal ability, and influences the amount and composition of resources used | Continuous | See Wetherbee et al. 2020a, Appendix II |
| Wood diameter preference | Diversity of preferred habitats may aid the decomposition process | Continuous | Gossner et al. 2013, Seibold et al. 2014, Janssen et al. 2017 |
| Decay stage preference | Diversity of preferred habitats may aid the decomposition process | Continuous | Gossner et al. 2013, Seibold et al. 2014, Janssen et al. 2017 |

**Table 2. Variables that describe the oak trees and their surrounding landscape in southern Norway.**

| Experimental variables | Type | Measurement | Reference |
|---|---|---|---|
| Type of tree | Categorical (2 levels) | Veteran or young tree: a veteran tree was defined as a tree of at least 200 cm circumference with a visible cavity in the trunk, and a young tree was defined as having a circumference less 200 cm, and not having a visible hollow if it was larger than 95 cm | Lovdata (2011) |
| Location of the wood bundle | Categorical (2 levels) | Ground or canopy: bundles were either placed on the ground or hung from a branch in the middle of the canopy | Seibold et al. (2018) |
| **Additional variables** | | | |
| Habitat class | Categorical (2 levels) | Forest or open landscape: based on 50 m radius surrounding the tree. Open landscapes were either parks or agricultural landscapes | Sverdrup-Thygeson et al. (2010) |
| Tree cover density (2 variables) | Continuous | Measured at two scales (20 and 100 m): the 20 m scale was measured as the percent of the 20 m pixel where focal tree is located that is covered by forest. The 100 m scale was measured as the percent of 20 m pixels covered by forests within 100 m radius of focal tree. | Copernicus Tree Cover Density (2012 & 15) |
| Tree circumference | Continuous | Tree circumference at breast height. | Sverdrup-Thygeson et al. (2010) |

These variables were used to predict species richness and functional diversity of beetles, as well as the decomposition rates of the experimentally added bundles of wood.

them with a linear mixed model with Gaussian distribution (LMM). Additionally, we modeled species richness (count data) with a Generalized linear mixed effect model with a Poisson distribution (GLMM) (see S2 Fig for variable distributions). We also included a random effect in each model with sampling block as a random intercept to deal with the spatial correlation introduced by the study design. Since decomposition is heavily influenced by abiotic conditions, we also tested if tree cover (Table 2) or moisture in the bundles differed between the two tree types. To calculate the tree cover, we used we used Copernicus tree cover density maps with 20 m resolution and measured the tree cover density within a 20 m and a 100 m radius of the focal tree [57].

Prior to statistical analysis, we followed the steps for data exploration outlined by Zuur et al. (2010) for all statistical models. The best model in each case was chosen with backward model selection based on Akaike information criterion (AIC). We subsequently visually checked the assumptions of the final LMMs of normal distribution of the residuals and homoscedasticity [58]. We also checked the final GLMM for over/under dispersion with the function *dispersion_glmer* from the 'blmeco' package [59]. Additionally, we checked for influential observations, and spatial and temporal structure that was not accounted for by the model, by plotting the model residuals against the spatial coordinates and looking for patterns [58]. The following packages were also used for data manipulation, statistical analysis and graphical visualization: 'lattice' [60], 'ggplot2' [61], and 'dplyr' [62].

## Results

Over the course of the two summers we captured a total of 465 beetle species (4,539 individuals) in the flight intercept traps, of which 160 (1,405 individuals) were wood decomposers (S1 Table). The total number of wood-decomposing beetle species captured around veteran trees was 132 (787 individuals), compared to 114 (618 individuals) around young trees. Both species richness and functional diversity of wood-decomposing beetles were higher around veteran trees than young trees (Fig 1 and Table 3, P<0.001 and P = 0.021, respectively). This effect was especially pronounced for species richness, where there were on average 22 (min = 10, max = 24) species around veteran trees and 16 (10, 29) around young trees. The community weighed mean (CWM) of wood diameter preference for beetles captured around the veteran trees was significantly higher than for those captured around the young trees (P = 0.011),

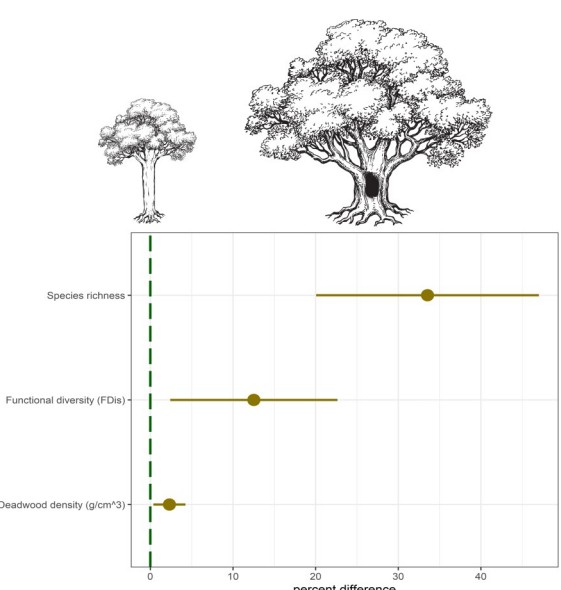

**Fig 1. The percent difference between paired young (green line) and veteran (brown points) oak trees for wood-decomposing beetle species richness, beetle community functional diversity, and the post-experiment density of the experimentally added dead wood.** Error bars show 95% confidence intervals, based on regression models (see Table 3 for model outputs). All sites were located in southern Norway.

**Table 3. Estimated regression parameters, standard errors and P-values from the best models.**

| Response variable and predictors | | Estimate | Standard error | P value | $R^2$ (or pseudo $R^2$) |
|---|---|---|---|---|---|
| **Species Richness** | | | | | |
| intercept | | 2.746 | 0.086 | <0.001 | 0.18 |
| Type of tree | (veteran) | 0.289 | 0.074 | <0.001 | |
| **FDis** | | | | | |
| intercept | | 1.293 | 0.047 | <0.001 | 0.13 |
| Type of tree | (veteran) | 0.162 | 0.067 | 0.021 | |
| **CWM wood diameter** | | | | | |
| intercept | | 2.202 | 0.047 | <0.001 | 0.15 |
| Type of tree | (veteran) | 0.180 | 0.067 | 0.012 | |
| **Wood density** | | | | | |
| intercept | | 0.640 | 0.005 | <0.001 | 0.16 |
| Type of tree | (veteran) | 0.015 | 0.006 | 0.023 | |
| Surroundings | (forest) | 0.019 | 0.007 | 0.007 | |
| **Percent weight loss** | | | | | |
| intercept | | 0.482 | 0.003 | <0.001 | 0.16 |
| Type of tree | (veteran) | -0.009 | 0.003 | 0.011 | |
| Surroundings | (forest) | -0.010 | 0.004 | 0.03 | |

The intercept represents young oaks or, in models with a 'surroundings' covariate, young trees in open landscapes. Species richness of wood decomposing beetles was modeled with a GLMM with Poisson distribution (N = 40). Beetle functional diversity (FDis) and community weighted mean (CWM) of their wood diameter preference were modeled with LMM with Gaussian distribution (N = 40), as was the density of the experimentally added bundles of wood (N = 80). The best models were identified with backward model selection based on the AIC. Note: the model estimates for species richness have not been back transformed. Results from additional models that did not have significant effects are presented in text.

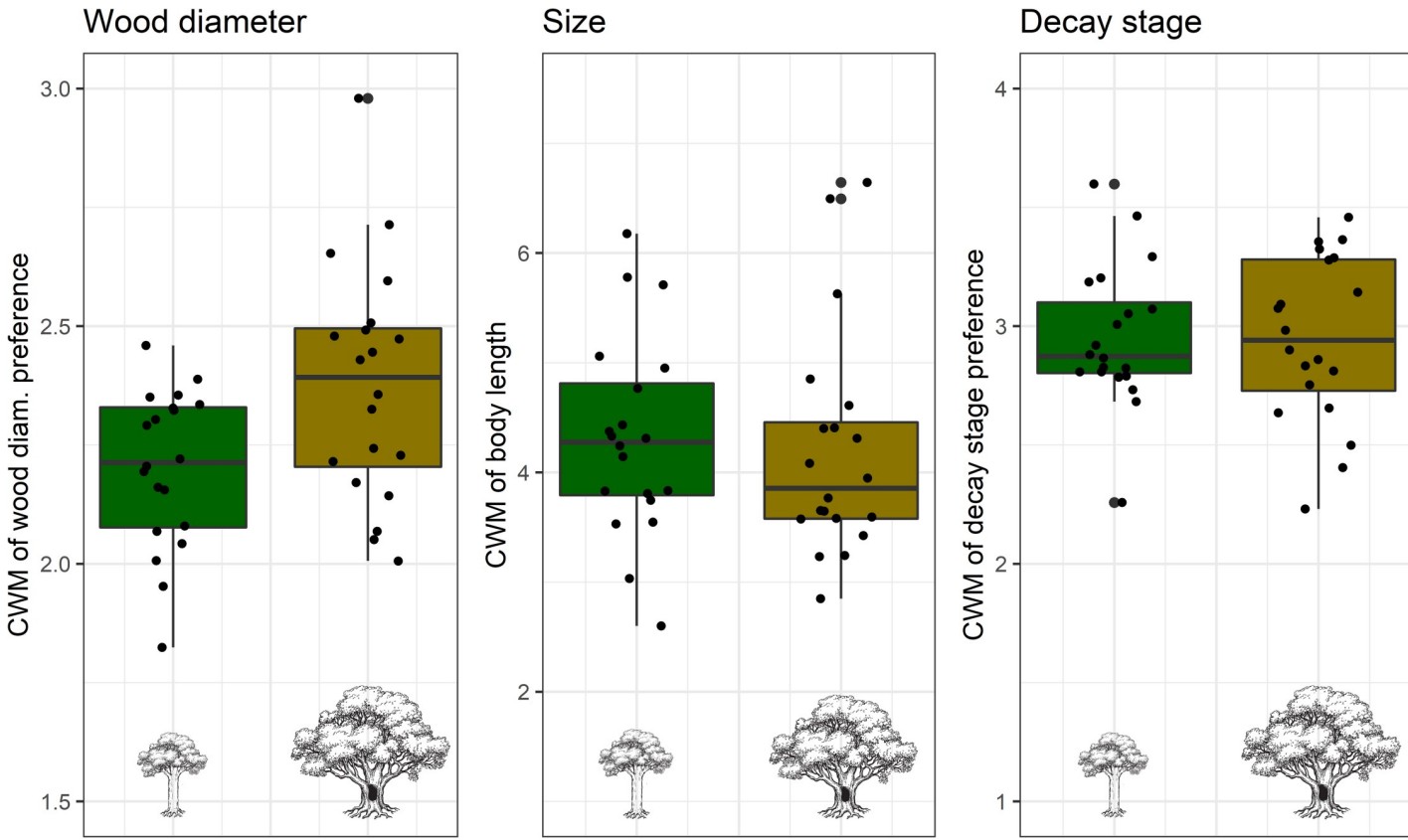

**Fig 2. Community weighted mean (CWM) of the three traits that we included in the measure of functional diversity for wood decomposing beetles captured around young (green) and veteran (brown) oak trees in southern Norway.** The plots show the median, first and third quartiles, with whiskers that extend 1.5 times the interquartile range. All observations as shown as points on the plots. Beetle communities around veteran trees preferred significantly larger diameter wood (middle plot) than those around young trees (P = 0.01), whereas the CWM of decay stage preference (left plot) and beetle body length (right plot) did not differ among tree types.

whereas the CWM of body length and wood decay stage preference did not differ between the two tree types (Fig 2).

The diversity and abundance of beetles reared from the experimental wood bundles was low (19 individuals from five species), but most of the species and individuals came from bundles that had been placed around veteran trees (Table 4). However, the type of tree (veteran or young) did influence the final density of the wood and the percent weight loss of the bundles

**Table 4. Species and abundances of beetles extracted from the experimentally added bundles of small diameter wood.**

| Species | Veteran tree abundance | Young tree abundance |
|---|---|---|
| *Orchestes fagi* | 1 | 0 |
| *Phymatodes testaceus* | 1 | 0 |
| *Poecilium alni* | 0 | 2 |
| *Salpingus planirostris* | 2 | 0 |
| *Scolytus intricatus* | 13 | 0 |

The bundles of wood were placed around either a veteran or a young tree and left in the field for two seasons and then extracted for one year. The study was carried out in southern Norway.

(PWL). After two seasons in the field, the density of the experimentally added wood was, on average, 2.3% higher in the bundles placed around the veteran trees than those around young trees, and this effect was mirrored in the PWL, which was significantly lower around the veterans (Fig 1 and Table 4, P = 0.023 and P = 0.011, respectively). Independent of the type of tree, wood density was 2.9% higher in forests than in open landscapes and again this result was mirrored in the PWL, which was significantly lower in forests (Table 3, P = 0.007 and P = 0.03, respectively). We also found that there were no significant differences between the veteran and young trees regarding tree cover density, or the amount of moisture in the bundles at the end of the study (S2 Table, P = 0.484 and 0.166, respectively).

## Discussion

Overall, the number of beetle species involved in decomposition of wood was higher around the veteran oaks than the young oaks. This finding is consistent with previous research in Germany, which found higher species richness of saproxylic beetles around veteran trees [1], and provides reasons to protect veteran trees as valuable habitat for beetle diversity. Our results indicate that veteran trees contribute greatly to a landscape's biodiversity. We observed that beetle species richness was more than thirty percent higher around veteran oaks than around nearby (only 50 to 200 meters) young oaks in the same landscape context. This pattern was mirrored by a less pronounced, yet still significant, increase in functional diversity around veteran oaks.

The number of beetle species that we collected in this study, although high, is in line with other observations around veteran oaks in Norway [48, 63–65], and a significant subset of these species appear to specialize on oaks [66]. Prior studies, however, have focused exclusively on veteran oaks and this is, to our knowledge, the first study to measure the diversity of beetles around young oaks in Norway. Although we captured many species around the young oaks, these numbers were likely higher due to the presence of veteran trees in the area. Research indicates that the number of veteran trees in an area (at scales up to 10 km) is a positive predictor of beetle diversity [48, 67, 68].

As our measure of functional diversity indicated, the species captured around veteran trees had greater differences in traits than those captured around the young trees. The main difference we detected in beetle functional diversity was that the beetles around veteran trees included species that preferred larger diameter dead wood, while fewer such species occurred around the young trees. Examples of these beetles were *Ptinus subpillosus*, *Cryptophagus micaceus*, *Dorcatoma chrysomelina*, and *Euglenes oculatus*. Of these, *E. oculatus* is on the Norwegian red list as near threated. Also, *C. micaceus* and *D. chrysomelina* are primarily fungivores, and this highlights the importance of fungal fruiting bodies that are associated with large diameter wood as a habitat provided by veteran trees.

Additionally, several species of bark beetle (family: *Curculionidae*) were observed in higher numbers around the veteran trees than the young trees, and this finding has implications for colonization of small diameter wood that we added around the trees. Specifically, *Scolytus intricatus* was observed in higher numbers around the veteran trees. The species is known to have a strong oak association [69], and is specialized in small diameter wood in early decay [70]. This seems to be supported by our data, as most of the beetles extracted from the bundles were *S. intricatus*. However, we only extracted 13 individuals from five different bundles. In fact, there were surprisingly few beetles extracted from the bundles, especially considering how many beetles we captured in the flight intercept traps. This may be explained by the fact that *S. intricatus*, like most bark beetles, colonize wood in June and overwinter as larvae, then emerges the following summer [69]. In this study we left the bundles in the field for two seasons to

measure decomposition over a longer period, which may have reduced the number of beetle that we captured in the extractions.

In general, previous research has found high abundance and diversity of beetles in small diameter wood [32, 33, 35]. Our results contrast greatly with these findings, but is more in line with Ferro and Gimmel (2014), who found much lower colonization rates. To some extent the differences between the findings may result from differences in the amount and diversity of dead wood that was placed out in the various studies, as this is important for saproxylic species richness and abundance [26, 71–73]. Additionally, beetles colonizing small diameter wood are known to be sensitive to freshness of the wood and seasonality [74], and many emerge as adults after one year [32, 33]. As mentioned above, it is quite likely that since the bundles were in the field for two seasons, we missed the early emerging beetles. Also, freshness of the sticks may have played a role in the results, as the sticks were not placed out immediately after cutting. It is less likely that seasonality was important, because the bundles were placed out in spring before the peak flight activity of most beetles in Norway [75].

In addition to few beetles being extracted from the bundles, we found that the bundles placed around the young trees had lower density and greater weight loss than those placed around the veteran trees. This indicates that, contrary to what we had expected, the higher levels of beetle diversity that we observed around the veteran oaks did not increase decomposition rates. While functional dispersion has been shown to be an important measure of trait diversity for decomposition [12], this of course depends on what traits are used to estimate this measure of functional diversity [76]. As discussed above, we found that the main difference in beetle functional diversity between the tree types was that the beetles around veteran trees preferred larger diameter dead wood. Since we only measured decomposition of small diameter wood, it is perhaps not surprising that functional diversity of beetles was not positively correlated with decomposition rates in our study.

However, this does not explain why we observed lower decomposition rates of bundles placed around veteran trees. Lower decomposition rates are unlikely to be attributed to differences in abiotic conditions since we found no differences in tree cover density at the 20 m and 100 m scale, or in the amount of moisture in the wood bundles, between the veteran and young trees. One possible explanation for this finding is related to differences in fungal diversity between the two tree types. High levels of fungal diversity have been shown to slow decomposition rates [23, 27, 77]. Fungal diversity also increases with the diameter of the tree [25], and veteran trees have more fungal fruiting bodies than typical forest trees [2, 78]. It has also been found that beetles may act as targeted dispersers of fungal spores even when not directly colonizing the dead wood themselves [40]. It is therefore possible that more fungal spores arrived, directly or indirectly via beetle dispersal, in the bundles of wood placed around the veteran trees, and that this slowed decomposition. Unfortunately, we did not measure fungal diversity; an experimental design with a series of beetle and fungi exclusions could be used to verify these findings and gain a better understanding of the underlying mechanisms.

## Conclusion

Our results highlight that veteran trees have a high conservation value and have species-rich beetle communities with high functional diversity. This has often been assumed but rarely measured, and these results provide hard evidence of the benefits that arise from protecting veteran trees. Our results also indicate that the presence of veteran trees is linked to slower decomposition rates of small diameter wood during early decay. The mechanism behind this finding remains unknown, but could potentially be caused by higher fungal diversity, which has been linked to slower wood decomposition rates. Veteran trees provide an ecological

legacy within anthropogenic landscapes that influence ecosystem functions and services. Actions to protect veteran trees are urgently needed in order to save these valuable organisms and their associated biodiversity.

## Supporting information

**S1 Fig. Correlation matrix with continuous variables used in the analysis of beetle diversity and wood decomposition rates around oaks in Southern Norway.** Values on top right are the Pearson's correlation coefficient.
(DOCX)

**S2 Fig. Distribution of values from beetle sampling and experimentally added wood bundles around oaks in southern Norway.** Histograms show beetle species richness, density of wood bundles, functional diversity (FDis), community weighted mean (CWM) of species' wood decay stage preference, species' wood diameter preference, and beetle body length.
(DOCX)

**S1 Table. Beetle species captured in flight intercept traps in veteran (VT) and young (YT) trees and characterized as wood-decomposers.** Beetles were classified as being wood decomposers based on being involved in primary or secondary wood decomposition at any point in their life stages. This included the following feeding types (FT): xylophagous (x), mycetophagous (m), saprophagous (s) and polyphagous (p). Additionally, the trait data regarding the beetle's body length (mm), wood diameter preference (WD pref) and wood decay stage preference (D pref), and references for their feeding type is provided.
(DOCX)

**S2 Table. Estimated regression parameters, standard errors and P-values from models that predicted the starting wet weight and density of the bundles, bundle wetness after the experiment, and Tree Cover Density (TCD) at 20 m and 100 m scales.** All models compare values between veteran and young oak trees. All response variables were modeled with LMMs with Gaussian distribution.
(DOCX)

## Acknowledgments

We would like to thank Sarah DeGennaro, Johan Kjorven and Alexius Folk for assistance in the field, Sindre Ligaard for identifying the beetles, Irmelin Gram-Hanssen for proofreading the manuscript and Matthew Cooper for making the illustrations. We would also like to thank the landowners for allowing permission to access their land; without their support this research would not have been possible.

## Author Contributions

**Conceptualization:** Ross Wetherbee, Tone Birkemoe, Anne Sverdrup-Thygeson.

**Data curation:** Ross Wetherbee.

**Formal analysis:** Ross Wetherbee, Ryan C. Burner.

**Funding acquisition:** Tone Birkemoe, Anne Sverdrup-Thygeson.

**Investigation:** Ross Wetherbee.

**Methodology:** Ross Wetherbee, Anne Sverdrup-Thygeson.

**Project administration:** Tone Birkemoe, Anne Sverdrup-Thygeson.

**Supervision:** Tone Birkemoe, Ryan C. Burner, Anne Sverdrup-Thygeson.

**Visualization:** Ross Wetherbee.

**Writing – original draft:** Ross Wetherbee.

**Writing – review & editing:** Tone Birkemoe, Ryan C. Burner, Anne Sverdrup-Thygeson.

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
