## [Decision Letter · Decision Letter 0]

14 Jan 2021

PONE-D-20-39244

Veteran trees have divergent effects on beetle diversity and wood decomposition

PLOS ONE

Dear Dr. Wetherbee,

Thank you for submitting your manuscript to PLOS ONE. After careful consideration, we feel that it has merit but does not fully meet PLOS ONE’s publication criteria as it currently stands. Therefore, we invite you to submit a revised version of the manuscript that addresses the points raised during the review process.

The reviewers raise legitimate points and I agree with these.  Please pay close attention to their concerns and modify the manuscript accordingly.  In addition to comments identified by reviewers, I found the description of the statistical analyses to be insufficient and lacking clarity. 

Generally, a manuscript requiring major revision will undergo another round of review after the modified version is resubmitted.

We look forward to receiving your revised manuscript.

Kind regards,

Janice L. Bossart

Academic Editor

PLOS ONE

Journal Requirements:

Reviewers' comments:

Reviewer's Responses to Questions

**Comments to the Author**

1. Is the manuscript technically sound, and do the data support the conclusions?

Reviewer #1: Partly

Reviewer #2: Partly

2. Has the statistical analysis been performed appropriately and rigorously? 

Reviewer #1: Yes

Reviewer #2: Yes

3. Have the authors made all data underlying the findings in their manuscript fully available?

Reviewer #1: No

Reviewer #2: No

4. Is the manuscript presented in an intelligible fashion and written in standard English?

Reviewer #1: Yes

Reviewer #2: Yes

5. Review Comments to the Author

Reviewer #1: This article is a useful contribution to understanding the role of decomposing wood in the biodiversity profile of forests and provides a strong defense of preserving and enhancing dead wood resources in those habitats. The methods are straightforward and clearly stated, but the presentation of discussion and comparisons with results of other studies in temperate forest systems could be improved. The authors fail to mention a significant body of work by Ferro et al. that was conducted in North America. Those studies are published as open access in the journal Insecta Mundi and should be consulted and considered in the discussion where results of previous work are compared. Further, the identification of 667 species must represent a substantial percentage of the total beetle diversity known from Norway, so a more general comparison of how the results relate to the overall diversity of beetles in the region would improve the discussion. Total beetle diversity in Norway is reasonably well documented and the data should be accessible. I answered “no” to the data availability question because the availability of the taxon dataset was not specifically addressed. A simple statement in the results would address this.

The methods used to conduct the taxonomic work are not clearly explained, so the reliability of the species list cannot be assessed. If the taxonomic work was conducted based on authoritative knowledge of the individual(s) who did the work, that should be stated. Otherwise references to keys, reference collections, and/or other identification tools should be described. I note that a single “expert” is credited with the identifications and presumably sorting of the 600+ beetle species and nearly 10,000 individuals. Why this person is not included as a co-author is beyond me, and perhaps this should be considered. Does the taxonomic work only count for a mention in the acknowledgements even though the entire paper rests on this dataset?

Reviewer #2: [I received no “supplemental data” and “supplemental data” are not referred to in the MS, therefore I assume a species list doesn't exist. If it does, then many remarks below can be ignored.]

The MS represents two studies. First a comparison of saproxylic beetle species collected in the canopy of veteran vs. young oaks. Second a comparison of twig bundles (1. colonization by beetles and 2. changes in densities) when influenced by the presence of a veteran or young oak.

The canopy study, alone, is of great interest and should be published, but MUST be rewritten and added to. As presented, all the MS provides are two vague proclamations: 1) “the number of beetle species involved in decomposition of wood was higher around the veteran oaks than the young oaks”; and 2) “beetle communities around the veteran trees had a greater diversity of traits relevant for decomposition of wood than those around young trees”. Which of the 170 species drove those differences? Were there any species that indicated or perhaps required veteran trees, or young trees? If we are interested in veteran trees for conservation purposes, then it would be good to know something about which species they help conserve.

A table, listing the 170 species, and how many were collected at each tree type, would provide future researchers with an enormous amount of useable data and greatly enhance the quality of the research. (Presumably, you already have created such a table for your analysis.) For example, someone looking into the habits of Species X or Genus Y could now use your observations. [Analyzing those 170 species would enhance the MS, or represent another paper entirely.]

The “Wood Decomposer” trait in Table 1 that came from “Diverse literature” represents witchcraft at this point. There is no way for anyone to know or understand how or why you designated a species the way you did. In the recommended table above, you should add a citation and tell which decomposer type you assigned to each species.

The twig study should be removed from this MS. It should be rewritten as a Note that stands as an example of negative results showing that the number of twigs was inadequate. Based on this study and other studies, an estimate of the required number of twigs to get meaningful results could be given. The differences in density could also be mentioned as interesting and worthy of further study, but beyond that, NO conclusions can be drawn. Small sample size, no controls, and no measure of fungal diversity in the area or in the twigs was taken.

**** Despite all my complaints, it’s a good study and please do publish it. Toss out the twigs and provide a species list and your paper will be read and cited long into the future. ;)

Below are some notes I jotted down as I read the MS.

Why would beetles colonizing fresh twigs be influenced by a veteran tree? Presumably fresh broken twigs are available throughout the forest, while veteran trees contain rare types of really rotten dead wood (“veteris” wood), see Ulyshen 2018. Different beetles go to different wood decay stages and sizes.

There is very little literature on beetles in fine wood debris, twigs, etc. Ferro and Vogel have published some papers on it. See below.

The time of year when small twigs are cut and placed to attract saproxylic beetles is important. See Ferro and Gimmel 2014. At the very least, specific dates of cutting and placement should be included.

Concerning the twig bundles, in relation to the dry weight and density measure, there were no controls in the experiment. Control 1: analysis of the twigs immediately after they are cut to establish a baseline (weight means nothing, you didn’t dry it); Control 2: bundles of twigs where beetles are excluded (perhaps with netting); Control 3: where fungi were excluded (perhaps the twigs were soaked in an antifungal agent prior to placement); Control 4: fungi and beetles excluded. You could even include a Control 5 where “natural weathering” is reduced by placing twigs in an environment (just on a shelf in the lab) where temp, humidity, etc. are held constant. It’s not your fault, I’ve seen virtually no controls used in saproxylic research.

If the twigs were fresh cut, the species that arrive at the twigs prefer fresh material. If those species only require one year to develop, those will be gone before rearing. Species that arrive the second summer are probably attracted to dead, drier wood, and are probably less numerous. What species did you get? Without a list, no one can tell if this is the case.

Where is the species list? You went to all that trouble to collect all those specimens, someone spent how many hours IDing them, and all that information is lost? All you got out of that effort were a couple means, P values, and a graph? Were the specimens destroyed or retained? If retained, where? How would a future researcher check your work or recalculate your findings? How could anyone compare or combine these findings to future studies?

“We also tested if abiotic conditions known to influence decay rates were different between the two tree types.” “We also found that abiotic conditions related to sun exposure and moisture content…” By analyzing the species collected? If you took abiotic measurements, temperature, humidity, etc., you need to say in the MM how and when you did that, and how you analyzed it, provide some citations to show how those conditions do influence decay rates, and give results.

Table 1: How do the traits of adult beetles have anything to do with the wood they inhabit as larvae (arguably the most important stage)? “Diverse literature” is meaningless and can’t be reproduced or reanalyzed.

Vogel, S. et al. 2020. Diversity and conservation of saproxylic beetles in 42 European tree species: an experimental approach using early successional stages of branches. Insect Conservation and Diversity. doi: 10.1111/icad.12442

Ulyshen, M. D. 2018. Saproxylic Insects. Springer Nature. Zoological Monographs I: 1–904.

Ferro, M. L., M. L. Gimmel, K. E. Harms, and C. E. Carlton. 2009. The beetle community of small oak twigs in Louisiana, with a literature review of Coleoptera from fine woody debris. The Coleopterists Bulletin 63: 239–263.

Ferro, M. L., and M. L. Gimmel. 2014. Season of fine woody debris death affects colonization of saproxylic Coleoptera. The Coleopterists Bulletin 68(4): 681–685.

6. PLOS authors have the option to publish the peer review history of their article (what does this mean?). If published, this will include your full peer review and any attached files.

Reviewer #1: No

Reviewer #2: No

---

## [Author Response · Author response to Decision Letter 0]

12 Feb 2021

Comments to the Author

Editor: The reviewers raise legitimate points and I agree with these. Please pay close attention to their concerns and modify the manuscript accordingly. In addition to comments identified by reviewers, I found the description of the statistical analyses to be insufficient and lacking clarity. 

Our response: We appreciate the comments and have tried to explain our analysis more thoroughly (lines: 181 to 232). We also included a supplementary figure with distribution of the response variables (Fig S2) and a correlation matrix with the predictor variables (Fig S1). 

1. Is the manuscript technically sound, and do the data support the conclusions?

Reviewer #1: Partly

Reviewer #2: Partly

Our response: Both reviewers have requested a table with species and their abundances collected around the two tree types be included in the manuscript. We agree that this was needed and Table S1 (all wood-decomposing beetles) and Table 4 (beetles extracted from the wood bundles) are added to increase transparency. All data regarding species occurrences have been registered in Global Biodiversity Information Facility (GBIF) and is publicly available (https://doi.org/10.15468/5bxyph). This reference was accidentally omitted from the original manuscript and is now included.

2. Has the statistical analysis been performed appropriately and rigorously? 

Reviewer #1: Yes

Reviewer #2: Yes

3. Have the authors made all data underlying the findings in their manuscript fully available?

The journal requires authors to make all data underlying the findings described in their manuscript fully available without restriction, with rare exception (please refer to the Data Availability Statement in the manuscript PDF file). The data should be provided as part of the manuscript or its supporting information, or deposited to a public repository. For example, in addition to summary statistics, the data points behind means, medians and variance measures should be available. If there are restrictions on publicly sharing data—e.g. participant privacy or use of data from a third party—those must be specified.

Reviewer #1: No

Reviewer #2: No

Our response: We have added the data that the reviewers requested (Tables S1 and 4), and all data regarding species observations are made available on GBIF (https://doi.org/10.15468/5bxyph). We will also archive all data in NMBU Open Research Data (https://dataverse.no/dataverse/nmbu).

 4. Is the manuscript presented in an intelligible fashion and written in standard English?

Reviewer #1: Yes

Reviewer #2: Yes

 

Reviewer #1: This article is a useful contribution to understanding the role of decomposing wood in the biodiversity profile of forests and provides a strong defense of preserving and enhancing dead wood resources in those habitats. The methods are straightforward and clearly stated, but the presentation of discussion and comparisons with results of other studies in temperate forest systems could be improved. 

1.) The authors fail to mention a significant body of work by Ferro et al. that was conducted in North America. Those studies are published as open access in the journal Insecta Mundi and should be consulted and considered in the discussion where results of previous work are compared. 

Our response: Thank you for the suggested literature. The work by Ferro et al. is very relevant and we now discuss it in both the introduction (lines: 77 to 103) and the discussion (lines: 339 to 353).

2.) Further, the identification of 667 species must represent a substantial percentage of the total beetle diversity known from Norway, so a more general comparison of how the results relate to the overall diversity of beetles in the region would improve the discussion. Total beetle diversity in Norway is reasonably well documented and the data should be accessible. I answered “no” to the data availability question because the availability of the taxon dataset was not specifically addressed. A simple statement in the results woul¬¬¬¬¬¬¬¬¬d address this.

Our response: We unfortunately reported too many total beetle species. Sorry for the mistake. The correct number (465 species) is now included in the manuscript. Also, a total of 10 species had to be removed from our analysis of wood-decomposing beetles (now 160 species). These changes had little effect on our findings overall. The total number of registered beetle species in Norway is approximately 3600. Thus, our catch makes up approximately 13%. However, as this study is carried out in one of the most productive and species rich areas, this high number was expected and has been found in a larger number of studies of hollow oaks in the region (see literature listed in the manuscript, lines: 306 to 315). Thus, we do not think a discussion of overall species richness relating to the region improves our study. Species occurrences were registered in GBIF but the reference was accidently omitted in the first draft (https://doi.org/10.15468/5bxyph). In addition, we have added a table of the decomposing beetles caught around veteran and young trees and the beetles extracted from the wood bundles (Table S1 and Table 4). We apologize for omitting this previously. 

3.) The methods used to conduct the taxonomic work are not clearly explained, so the reliability of the species list cannot be assessed. If the taxonomic work was conducted based on authoritative knowledge of the individual(s) who did the work, that should be stated. Otherwise references to keys, reference collections, and/or other identification tools should be described. I note that a single “expert” is credited with the identifications and presumably sorting of the 600+ beetle species and nearly 10,000 individuals. Why this person is not included as a co-author is beyond me, and perhaps this should be considered. Does the taxonomic work only count for a mention in the acknowledgements even though the entire paper rests on this dataset?

Our response: Our expert, Sindre Ligaard, would certainly have the right to co-authorship, but he has chosen not to coauthor publications resulting from his taxonomic work and we respect his wishes. He is of course credited for his work on the dataset in the GBIF registry (https://doi.org/10.15468/5bxyph). We have also included a reference to the taxonomy used in the manuscript as requested: https://www.biodiversity.no/ScientificName/Insecta/89

Reviewer #2: Despite all my complaints, it’s a good study and please do publish it. Toss out the twigs and provide a species list and your paper will be read and cited long into the future. There is very little literature on beetles in fine wood debris, twigs, etc. Ferro and Vogel have published some papers on it. See below.

1.) I received no “supplemental data” and “supplemental data” are not referred to in the MS, therefore I assume a species list doesn't exist. If it does, then many remarks below can be ignored.

Our response: Our species list with abundances was published on GBIF (https://doi.org/10.15468/5bxyph) but was accidentally omitted in the manuscript. Additionally, we have now included a table of the beetle species caught around veteran and young trees and those extracted from the bundles of wood (Table S1 and Table 4).

2.) The MS represents two studies. First a comparison of saproxylic beetle species collected in the canopy of veteran vs. young oaks. Second a comparison of twig bundles (1. colonization by beetles and 2. changes in densities) when influenced by the presence of a veteran or young oak.

Our response: We disagree with this statement. Sampling happened in parallel and methods were centered around a central hypothesis: decomposer beetle diversity will be higher around veteran trees and increase decomposition rates correspondingly.

3.) The canopy study, alone, is of great interest and should be published, but MUST be rewritten and added to. As presented, all the MS provides are two vague proclamations: 1) “the number of beetle species involved in decomposition of wood was higher around the veteran oaks than the young oaks”; and 2) “beetle communities around the veteran trees had a greater diversity of traits relevant for decomposition of wood than those around young trees”. Which of the 170 species drove those differences? Were there any species that indicated or perhaps required veteran trees, or young trees? If we are interested in veteran trees for conservation purposes, then it would be good to know something about which species they help conserve.

Our response: We have highlighted the beetle groups driving the pattern in functional diversity between the veteran and young trees (discussion lines 316 to 324). This clearly contributes to explaining the patterns in functional diversity between the two tree types.

4.) A table, listing the 170 species, and how many were collected at each tree type, would provide future researchers with an enormous amount of useable data and greatly enhance the quality of the research. (Presumably, you already have created such a table for your analysis.) For example, someone looking into the habits of Species X or Genus Y could now use your observations. [Analyzing those 170 species would enhance the MS, or represent another paper entirely.

Our response: Thank you for your comment, the requested information is now included in Table S1.

5.) The “Wood Decomposer” trait in Table 1 that came from “Diverse literature” represents witchcraft at this point. There is no way for anyone to know or understand how or why you designated a species the way you did. In the recommended table above, you should add a citation and tell which decomposer type you assigned to each species.

Our response: This have now been specified (lines 166 to 175) and a table with the species and assigned references can be found in the Supporting information (Table S1). 

6.) The twig study should be removed from this MS. It should be rewritten as a Note that stands as an example of negative results showing that the number of twigs was inadequate. Based on this study and other studies, an estimate of the required number of twigs to get meaningful results could be given. The differences in density could also be mentioned as interesting and worthy of further study, but beyond that, NO conclusions can be drawn. Small sample size, no controls, and no measure of fungal diversity in the area or in the twigs was taken.

Our response: The bundles of sticks ranged from 1 to 3cm in diameter and the average bundle weight was 1.89kg. In total we added 150kg of wood. Of course, it would be nice to have more observations, but 80 bundles are a reasonable number of observations in an experimental field study, considering that we were only testing the difference between the two tree types. Our study design allowed us to observe differences that were primarily the result of the tree. Reviewer 2 seems to accept that this was the case for the window traps, and it is also the case for the bundles of wood. As Reviewer 2 points out in their opening comment, very little has been done with beetles in small diameter dead wood, and as far as we are aware, nothing has been done relating this topic to veteran trees. Although this study had limitations which are discussed in the manuscript (lines: 339 to 353), we believe it adds to the body of work. We disagree that no conclusions can be drawn and believe it provides a valuable contribution to the understanding of how veteran trees and their communities contribute to ecosystem functions and services. However, it is true that we did not measure fungal diversity which would have provided highly valuable data and did not have a control for beetles or fungi. So, we have downplayed our conclusions regarding these findings in the abstract and discussion (366-381), but believe that this manuscript may encourage future research on the topic. 

7.) Why would beetles colonizing fresh twigs be influenced by a veteran tree? Presumably fresh broken twigs are available throughout the forest, while veteran trees contain rare types of really rotten dead wood (“veteris” wood), see Ulyshen 2018. Different beetles go to different wood decay stages and sizes.

Our response: In additional to other forms of dead wood, veteran trees have more dead wood in the canopy. This dead wood consists of both large and small diameter wood. Also, veteran trees have more species and greater numbers of individuals of saproxylic beetles trapped around them. Some of these species prefer small diameter wood and in early decay. Therefore, it is possible that small diameter wood would be colonized more around veteran trees, and this would result in faster decomposition. However, this has not be studied in any detail and this was one of our motivations for carrying out this study. Differences in preferences for wood decay stage and size is well documented, included in our measure of functional diversity and a central point in the manuscript. We have now discussed this in more detail (lines 316 to 324).

8.) The time of year when small twigs are cut and placed to attract saproxylic beetles is important. See Ferro and Gimmel 2014. At the very least, specific dates of cutting and placement should be included.

Our response: This is a good point and additional information has been added to the methods (lines: 142 to 152) and discussed (lines: 339 to 353). 

9.) Concerning the twig bundles, in relation to the dry weight and density measure, there were no controls in the experiment. Control 1: analysis of the twigs immediately after they are cut to establish a baseline (weight means nothing, you didn’t dry it); Control 2: bundles of twigs where beetles are excluded (perhaps with netting); Control 3: where fungi were excluded (perhaps the twigs were soaked in an antifungal agent prior to placement); Control 4: fungi and beetles excluded. You could even include a Control 5 where “natural weathering” is reduced by placing twigs in an environment (just on a shelf in the lab) where temp, humidity, etc. are held constant. It’s not your fault, I’ve seen virtually no controls used in saproxylic research.

Our response: We measured the weight of the bundles at the start of the experiment and since the sticks were all collected at the same time, from the same source, and randomized between the bundles this is a reasonable measure. We could not get the dry weight, because, as you have pointed out, this would decrease the quality of the bundles and possible influence colonization and decomposition. We also measured the density of the tips of each stick at the start of the experiment and found that there were no systematic differences (Supporting information, Table S2). Perhaps this was not entirely clear, so we have included the analysis of the percent weight loss of the bundles in the paper, which corresponds to the results regarding the density of the wood (Table 3).

The rest of this sounds like a nice follow up experiment and we have mentioned it in the discussion (lines: 379 to 381). The very point that you have designed a new experiment around our findings emphasizes their relevance. 

10.) If the twigs were fresh cut, the species that arrive at the twigs prefer fresh material. If those species only require one year to develop, those will be gone before rearing. Species that arrive the second summer are probably attracted to dead, drier wood, and are probably less numerous. What species did you get? Without a list, no one can tell if this is the case.

Our response: Yes, this is a good point and has been added to the discussion (lines: 339 to 353). See Table 4 for the species that were extracted from the bundles. 

11.) Where is the species list? You went to all that trouble to collect all those specimens, someone spent how many hours IDing them, and all that information is lost? All you got out of that effort were a couple means, P values, and a graph? Were the specimens destroyed or retained? If retained, where? How would a future researcher check your work or recalculate your findings? How could anyone compare or combine these findings to future studies?

Our response: We see your point about the species list, and it has been included (see comments above). 

12.) “We also tested if abiotic conditions known to influence decay rates were different between the two tree types.” “We also found that abiotic conditions related to sun exposure and moisture content…” By analyzing the species collected? If you took abiotic measurements, temperature, humidity, etc., you need to say in the MM how and when you did that, and how you analyzed it, provide some citations to show how those conditions do influence decay rates, and give results.

Our response: This should have been stated more clearly in the manuscript. We looked at tree cover density for sun exposure and bundle wetness as a measure of moisture at the sites. This has been clarified (lines 213 to 216, 284 to 287 and 368) and the results from that analysis have been included in the Supporting information (Table S2). 

13.) Table 1: How do the traits of adult beetles have anything to do with the wood they inhabit as larvae (arguably the most important stage)? “Diverse literature” is meaningless and can’t be reproduced or reanalyzed.

Our response: We have now clarified (Line 94 and Table 1): adult size is highly correlated with larvae size, and wood diameter and decay stage preference is the same for adults and larvae. What we had written previously was not clear and we have changed it. Diverse literature is removed and replaced with proper references (Table S1).

14.) Additional literature to include

Vogel, S. et al. 2020. Diversity and conservation of saproxylic beetles in 42 European tree species: an experimental approach using early successional stages of branches. Insect Conservation and Diversity. doi: 10.1111/icad.12442

Ulyshen, M. D. 2018. Saproxylic Insects. Springer Nature. Zoological Monographs I: 1–904.

Ferro, M. L., M. L. Gimmel, K. E. Harms, and C. E. Carlton. 2009. The beetle community of small oak twigs in Louisiana, with a literature review of Coleoptera from fine woody debris. The Coleopterists Bulletin 63: 239–263.

Ferro, M. L., and M. L. Gimmel. 2014. Season of fine woody debris death affects colonization of saproxylic Coleoptera. The Coleopterists Bulletin 68(4): 681–685.

Our response: Thank you for these suggestions. They have been added to both the introduction and discussion and greatly improve the manuscript.

---

## [Decision Letter · Decision Letter 1]

5 Mar 2021

Veteran trees have divergent effects on beetle diversity and wood decomposition

PONE-D-20-39244R1

Dear Dr. Wetherbee,

We’re pleased to inform you that your manuscript has been judged scientifically suitable for publication and will be formally accepted for publication once it meets all outstanding technical requirements.

Kind regards,

Janice L. Bossart

Academic Editor

PLOS ONE

Additional Editor Comments:

Although Reviewer 2 had no comments directed to the Author, they likewise felt the revisions were satisfactory.  Please carefully proofread your manuscript for any remaining errors and check that all citations are correct.

Reviewers' comments:

Reviewer's Responses to Questions

**Comments to the Author**

1. If the authors have adequately addressed your comments raised in a previous round of review and you feel that this manuscript is now acceptable for publication, you may indicate that here to bypass the “Comments to the Author” section, enter your conflict of interest statement in the “Confidential to Editor” section, and submit your "Accept" recommendation.

Reviewer #1: All comments have been addressed

Reviewer #2: (No Response)

2. Is the manuscript technically sound, and do the data support the conclusions?

Reviewer #1: Yes

Reviewer #2: (No Response)

3. Has the statistical analysis been performed appropriately and rigorously? 

Reviewer #1: Yes

Reviewer #2: (No Response)

4. Have the authors made all data underlying the findings in their manuscript fully available?

Reviewer #1: Yes

Reviewer #2: (No Response)

5. Is the manuscript presented in an intelligible fashion and written in standard English?

Reviewer #1: Yes

Reviewer #2: (No Response)

6. Review Comments to the Author

Reviewer #1: The authors have done a good job of addressing reviewer concerns and correcting some errors that were not obvious to reviewers in the original version. They still need to carefully proof the manuscript prior to final submission. The family name "Curculionidae" should not be in italics unless this is a quirk of journal requirements. There may be other proofing errors. I have no major criticisms of the revision.

Reviewer #2: (No Response)

7. PLOS authors have the option to publish the peer review history of their article (what does this mean?). If published, this will include your full peer review and any attached files.

Reviewer #1: No

Reviewer #2: No

---

## [Editor Report · Acceptance letter]

10 Mar 2021

PONE-D-20-39244R1 

Veteran trees have divergent effects on beetle diversity and wood decomposition 

Dear Dr. Wetherbee:

I'm pleased to inform you that your manuscript has been deemed suitable for publication in PLOS ONE. Congratulations! Your manuscript is now with our production department. 

Kind regards, 

on behalf of

Dr. Janice L. Bossart 

Academic Editor

PLOS ONE